# Vitamin C and Omega-3 Fatty Acid Intake Is Associated with Human Periodontitis—A Nested Case-Control Study

**DOI:** 10.3390/nu14091939

**Published:** 2022-05-05

**Authors:** Louisa Mewes, Carina Knappe, Christian Graetz, Juliane Wagner, Tobias J. Demetrowitsch, Julia Jensen-Kroll, Karim Mohamed Fawzy El-Sayed, Karin Schwarz, Christof E. Dörfer, Stefan Schreiber, Matthias Laudes, Dominik M. Schulte

**Affiliations:** 1Clinic for Conservative Dentistry and Periodontology, School of Dental Medicine, Kiel University, 24105 Kiel, Germany; louisa.mewes@charite.de (L.M.); graetz@konspar.uni-kiel.de (C.G.); karim.fawzy@gmail.com (K.M.F.E.-S.); doerfer@konspar.uni-kiel.de (C.E.D.); 2Department of Prosthodontics, Geriatric Dentistry and Craniomandibular Disorders, Charité—Universitätsmedizin Berlin, Corporate Member of Freie Universität Berlin, Humboldt-Universität zu Berlin, and Berlin Institute of Health, 14197 Berlin, Germany; 3Institute of Diabetes and Clinical Metabolic Research, University Hospital Schleswig-Holstein, Campus Kiel, 24105 Kiel, Germany; carina.knappe@uksh.de (C.K.); matthias.laudes@uksh.de (M.L.); 4Cluster of Excellence, Precision Medicine in Chronic Inflammation, Kiel University, 24105 Kiel, Germany; juliane.schulz@uksh.de (J.W.); stefan.schreiber@uksh.de (S.S.); 5Department of Oral and Maxillofacial Surgery, University Hospital of Schleswig-Holstein, Campus Kiel, 24105 Kiel, Germany; 6Department of Food Technology, Institute for Human Nutrition & Food Science, Kiel University, 24118 Kiel, Germany; tdemetrowitsch@foodtech.uni-kiel.de (T.J.D.); jjensenkroll@foodtech.uni-kiel.de (J.J.-K.); kschwarz-2@foodtech.uni-kiel.de (K.S.); 7Oral Medicine and Periodontology Department, Faculty of Dentistry, Cairo University, Giza 12613, Egypt; 8Department of Internal Medicine I, Division of Endocrinology, Diabetes and Clinical Nutrition, University Hospital Schleswig-Holstein, 24105 Kiel, Germany

**Keywords:** nutrition, periodontitis, vitamin C, omega-3 fatty acid, metabolite

## Abstract

Vitamins and omega-3 fatty acids (Ω3FA) modulate periodontitis-associated inflammatory processes. The aim of the current investigation was to evaluate associations of oral nutrient intake and corresponding serum metabolites with clinical severity of human periodontitis. Within the Food Chain Plus cohort, 373 periodontitis patients—245 without (POL) and 128 with tooth loss (PWL)—were matched to 373 controls based on sex, smoking habit, age and body mass index in a nested case-control design. The amount of oral intake of vitamins and Ω3FAs was assessed from nutritional data using a Food Frequency Questionnaire. Oral intake and circulatory bioavailability of vitamins and Ω3FA serum metabolomics were compared, using ultra-high-resolution mass spectrometry. Periodontitis patients exhibited a significantly higher oral intake of vitamin C and Ω3FA Docosapentaenoic acid (*p* < 0.05) compared to controls. Nutritional intake of vitamin C was higher in PWL, while the intake of Docosapentaenoic acid was increased in POL (*p* < 0.05) compared to controls. In accordance, serum levels of Docosapentaenoic acid were also increased in POL (*p* < 0.01) compared to controls. Vitamin C and the Ω3FA Docosapentaenoic acid might play a role in the pathophysiology of human periodontitis. Further studies on individualized nutritional intake and periodontitis progression and therapy are necessary.

## 1. Introduction

Periodontal disease is one of the most common chronic inflammatory diseases worldwide [1], characterized by an inflammatory progressive destruction of the tooth investing and supporting tissues, which untreated could ultimately lead to tooth loss [2].

A variety of factors have been linked to the etiology of periodontitis, including smoking, oral hygiene performance, genetic and epigenetic factors, systemic health and the patient’s nutritional status [3]. The most important behavioral factor linked to periodontitis remains routinely performed oral hygiene [4], and population-based interventions attempt to address behavioral factors to control periodontitis through legislation (antismoking, reduced sugar content in foods and drinks), restrictions (taxes on sugar and tobacco), guidelines and campaigns [5]. Although their preventive efficacy remains unclear, it seems indisputable that nutrition plays an important role in the course of systemic diseases [6]. It was postulated that oxidative stress promotes periodontal inflammation, and reduced levels of vitamin C [7,8,9,10] or increased levels of Ω6FAs [11,12] were associated with the greater severity of periodontitis and periodontal destruction. Although a systemic supplementation of vitamin C and E [13,14,15], D [16,17] or Ω3FAs [18] could reduce the risk of periodontitis and periodontal inflammation, or improve periodontal therapy outcomes, these effects seemed to wear off after a follow-up time of eight months [19].

Currently, the standardized periodontal therapy relies on manual supra- and subgingival debridement, with additional supportive antibiotic therapy under strictly defined clinical situations. However, once periodontitis is diagnosed, a common claim is to consider additional benefits of controlling the patients’ nutrition in combination with periodontal therapy [5]. The aims of the study were, therefore, (I) to evaluate the nutritional intake, especially of vitamins and fatty acids among a population with and without periodontal disease, (II) to investigate metabolite concentrations of candidate nutritional components resulting from (I), and (III) to look for associations between the nutritional profile and periodontal health.

## 2. Materials and Methods

### 2.1. Recruitment

Recruitment, phenotyping of dental status, questionnaire and matching periodontitis patients to controls were performed as published by Schulz et al. [20]. In brief, the nested case-control study was performed within the Kiel Food Chain Plus (FoCus) cohort. The FoCus cohort is located in the north of Germany, and 1837 participants were recruited between 2011 and 2014 [21,22]. While most participants were enrolled via the regional registration office, 511 subjects were recruited from the obesity outpatient clinic of the University Hospital of Schleswig-Holstein, Kiel, Germany. Subjects were extensively phenotyped including anthropometric measurements, nutritional habits and metabolomics. The overall aim of the FoCus cohort is to enable research into so-called metabolic inflammation. Parts of the questionnaire consisted of questions regarding dental status.

### 2.2. Phenotyping of Dental Status and Questionnaire

Phenotyping of dental status was performed via questionnaires as described before [20]. In brief, the participants were asked the following questions: Has periodontitis (gum disease) been diagnosed by a physician? If yes, in which year has periodontitis been diagnosed or how old were you? Have you been or are you undergoing dental treatment because of periodontitis? Did you lose teeth due to periodontitis (gum disease with progressed tooth loosening)? Participants were excluded from further analyses when statements were inconsistent. Only subjects who answered all of the questions negatively were accepted as potential controls [20].

### 2.3. Matching Periodontitis Patients to Controls

Clustering patients from the FoCus cohort according to their oral health status, 373 periodontally diseased patients were identified. These patients were further stratified into two groups: periodontitis patients without tooth loss (POL, *n*= 245) and periodontitis patients with tooth loss (PWL, *n* = 128). For comparability, 373 controls (COL, CWL) were matched (1:1) according to the gender, smoking, age and BMI of the periodontally diseased patients in a nested case-control design. The basic characteristics of the 373 identified periodontally diseased patients and their controls can be found in the manuscript of Schulz et al. [20].

### 2.4. Diet Acquisition

To record dietary habits, the scientifically recognized EPIC (European Prospective Investigation into Cancer and Nutrition) Nutrition Frequency Questionnaire, developed by the German Institute for Human Nutrition (DIfE) in Potsdam, was employed and has been validated, showing moderate, acceptable relative validity [23,24]. The questionnaire comprised specific questions on typical portion sizes and consumption frequency of the participants, who were asked to make retrospective statements about their previous year’s nutrition. Food intake frequencies and portion sizes were translated by the DIfE using EPIC-Soft, integrating the German food code (Bundeslebensmittelschlüssel, vII.3), into quantities of single nutrients in grams per day, e.g., a specific amino acid or specific polyunsaturated fatty acids.

### 2.5. Blood Sample Preparation

All blood samples were stored at −80 °C until the day of sample preparation. To avoid batch effects, all samples were randomly prepared and later measured. For sample preparation, an adapted version of the SIMPLEX-protocol from Matyash [25], a liquid–liquid extraction protocol, was used. From 100 µL blood samples, a lipophilic Methyl tert-butyl ether (MTBE) phase, a hydrophilic methanol-water phase and a protein pellet were obtained. All liquid phases were dried under vacuum (SpeedVac from Thermo Fisher, Henningsdorf, Germany) and resuspended with the following liquids: the lipophilic phase with isopropanol: chloroform (3:1) with 0.1% acetic acid; the hydrophilic phase with water/methanol (50/50, *v*/*v*) with 0.1% acetic acid. To each sample, 4 µL of an internal standard mixture was added, containing 13C-labeled tyrosine and tryptophan. The lipophilic samples were diluted in isopropanol/chloroform (3:1, *v*/*v*). The internal standard for these samples was synthetic lipids PC 5:0, PC 11:0, PC 19:0 and PG 17:0. All samples were then stored at −80 °C until the day of measurements.

### 2.6. Blood Sample Measurement and Data Evaluation

An FT-ICR-MS (7 Tesla, SolariXR, Bruker, Bremen, Germany) in the flow-injection mode (an HPLC 1260 Infinity from Agilent, Waldbronn, Germany) was used for the analyses. The eluent was water/methanol (50/50, *v*/*v*) with 0.1% acetic acid. The samples were ionized with an electrospray ionization source (in both modes) and a detection range of 65–950 *m*/*z*. The average resolution at 400 *m*/*z* was 600,000. The main instrument parameters were dry gas temperature (nitrogen) of 200 °C at 4 L min^−1^; nebulizer 1 bar; time-of-flight time section 0.35 ms and quadrupole mass 150 *m*/*z* with an RF frequency of 2 MHz, and detector sweep excitation power of 18%. The data evaluation was conducted with DataAnalysis 5.0 and MetaboScape 4.0.1 both from Bruker (Bremen, Germany). Sum formulas were calculated based on the mass error and isotopic fine structure. To reduce false-positive results, the seven golden rules of Kind and Fiehn were used [26]. In addition, annotation was conducted with customized databases by suspected targeted methods.

### 2.7. Statistical Analysis

The Statistical analyses and graphics were carried out with R [27]. A Shapiro–Wilk test was used to verify normality of the data. Wilcoxon–Mann–Whitney tests were used for group comparisons and *p*-values < 0.05 were considered significant. Multiple testing corrections for the subgroups of polyunsaturated fatty acids and vitamins of the nutritional data were performed using the Bonferroni method. Metabolite concentrations were detectable in only a limited number of subjects; therefore, instead of group comparisons, regression models were applied taking the matching factors (BMI, age, gender, smoking) as confounders into account. Metabolites below the limit of detection (1,000,000 counts) were set to NA. Metabolites were log-transformed. Logistic regression (C and P, C and POL, C and PWL) was performed using generalized linear models with family = binomial from the R stats package 3.4.4.

## 3. Results

The potential confounders (BMI, age, gender, smoking habits) served as matching criteria and were not significantly different between patients and controls (Table 1). Comparison of the two patient groups (POL vs. PWL) demonstrated significantly more smokers (63% vs. 75%, *p* = 0.038) and a higher age (56 (48–65) versus 63 (55–69) years, Median (IQR), *p* = 3.9 × 10^−5^) in the PWL group.

### 3.1. Daily Calorie Intake Showed No Difference between Periodontal Health and Disease

Food frequency questionnaire data were available from 652 subjects (326 sex-, smoker-, age- and BMI-matched pairs), 40% men and 65% smokers (Table 1). The average calorie intake of 9022 (7541–10,976) kJ/day in the periodontitis group (P) was not significantly higher than the daily calorie intake of their respective controls (C) with 8546 (7002–10,591) kJ/day (*p* = 0.054, Figure 1a). Furthermore, no significant differences in daily calorie intake were detected between POL and PWL compared to the pair-matched controls (COL, CWL) (*p* = 0.24; *p* = 0.11). Finally, no statistically significant difference in food intake in grams per day, independent of the energy content, could be observed overall (P/C), as well as between the groups POL/COL and PWL/CWL (Figure 1b).

### 3.2. Minerals, Protein, Carbohydrates and Fibers

No significant differences in the intake of minerals (*p* = 0.13), proteins (*p* = 0.08), carbohydrates (*p* = 0.26) and fibers (*p* = 0.34) between P and C could be demonstrated (Table 2).

### 3.3. Vitamin C Intake Is Increased in Subjects with Periodontitis

In general, P showed a significantly higher intake of organic acids in total compared to C (median difference: 9.24%, *p* = 0.02; Figure 2a), with no significant differences observed in the subgroups POL and PWL compared to pair-matched controls. In more detail, comparison of daily vitamin C intake suggested higher levels (median difference: 11.19%) in periodontally diseased patients compared to pair-matched controls (*p*_(nom)_ = 0.007, *p*_(corrected)_ = 0.091; Figure 2b). Here, a significantly elevated daily vitamin C intake in the PWL group (median difference: 11.48%) compared to CWL was detected (*p* = 0.022, Figure 2b). Additionally, vitamin D (*p*_(nom)_ = 0.033) exhibited an increase in its uptake in P compared to C, which cannot be found within the subgroups POL and PWL compared to their controls. No significant differences could be demonstrated for the intake of vitamin A, B1, B12, B2, B3, B5, B6, B7, B9, E and K (Table 2).

### 3.4. Fatty Acids

Overall, no significant differences were detected in the total uptake of fatty acids between P and C (*p* = 0.058), as well as for polyunsaturated- (*p* = 0.08; Figure 2c), monounsaturated- (*p* = 0.057) and saturated fatty acids (*p* = 0.86). However, more detailed analyses within the group of polyunsaturated fatty acids demonstrated a significantly higher intake of Docosapentaenoic acid (DPA, Ω3FA) in P compared to C (*p*_(nom)_ = 0.003, *p*_(corrected)_ = 0.036, Figure 2d). In detail, the intake of DPA was significantly higher in POL compared to their controls COL (*p*-value = 0.026, Figure 2d).

Further nominal significances comparing C and P were detected for the monounsaturated fatty acids Eicosenoic acid, Decosenoic acid and Tetracosenoic acid, as well as for the polyunsaturated fatty acids Docosahexaenoic acid, Eicodonic acid, Eicosatrienoic acid, Eicosatetraenoic acid, Octadecatetraenoic acid, Hexadecadienoic acid, Docosatetraenoic acid and Eicosadienoic acid intake. Results of all fatty acids tested are presented in Table 2. No further significances between C and P were detectable regarding the nutritional uptake of total Cholesterol (*p*_(nom)_ = 0.051) and Glycerol (*p*_(nom)_ = 0.054).

### 3.5. Serum Concentrations—Metabolome Data

To further investigate potential associations of the significant candidates from the nutrition questionnaire (vitamin C and DPA), we analyzed the corresponding metabolite concentrations. Comparing the intensities of all measured derivates of vitamin C (Ascorbic acid (Figure 3a, OR(CI) = 1.19 (0.74–1.90)), Dehydroascorbic acid, Threonate, and L-Gulonolactone), no significant differences could be observed regarding the subgroup analyses POL and PWL, and with respect to overall comparisons C/P with *p* = 0.35, *p* = 0.48, *p* = 0.99 and *p* = 0.50, respectively.

The earlier determined candidate fatty acid DPA was found to be higher in serum levels in P compared to C (*p* = 0.004, OR(CI) = 1.52 (1.15–2.01)), as well as within the subgroups POL and PWL (Figure 3b, OR(CI)_POL_= 1.39 (1.03–1.90), OR(CI)_PWL_ = 1.88 (1.22–2.94)). EPA was found to be higher in serum levels overall in P compared to C and in the subgroup POL compared to COL (Figure 3c, OR(CI) = 1.28 (1.04–1.58), OR(CI)_POL_ = 1.31 (1.03–1.67)).

## 4. Discussion

Periodontitis poses a worldwide health burden and is commonly associated with serious metabolic diseases, including type 2 diabetes mellitus (T2DM) and atherosclerosis [1,28]. T2DM and atherosclerosis are both highly connected to a misguided dietary intake and are proven to be positively influenced by conscious dietary regimes. Hence, nutritional factors could play an important role in a successful long-term medical therapy. A similar influence of a tailored diet in periodontitis could be of great interest, especially when associated with systemic ailments such as T2DM and atherosclerosis.

In the present study, we demonstrate for the first time a higher daily intake of vitamin C and DPA, an omega-3 fatty acid, in human periodontitis subjects. In detail, differences between the periodontitis groups (without and with tooth loss) compared to the matched controls (based on gender, smoking, age and BMI) could be demonstrated, where a higher vitamin C intake was associated with periodontitis patients with tooth loss and an elevated intake of DPA with periodontitis patients without tooth loss. However, from visual data inspection, we conclude that in the case of organic acids or DPA, sample size could have affected the statistical outcome, with the smaller subgroups unable to exceed significance thresholds while at the same time showing similar data distributions (Figure 2).

These findings seem to contradict those of previous studies on the effects of diet on periodontal disease [7,8,9,29], which demonstrated an elevated periodontitis risk associated with reduced vitamin C serum levels, malnutrition with a diet lacking Ω3FAs and vitamin C, and that periodontal health improvement could be achieved by their higher intake. To assess substrate uptake, these studies used serum levels of micronutrients (the substitution study method or a food frequency questionnaire). One cannot exclude the possibility that genetically determined maldigestion, malabsorption or increased metabolism could lead to significant differences between the amount of oral intake and corresponding serum levels. Not many studies compare the amount of oral intake, correspondingly measured serum levels and local availability of micronutrients, which would be essential for a completely objective comparison. In this context, the present study compared the specified amount of nutritional intake of vitamin C and DPA and its corresponding serum levels. Hereby, we were able to successfully verify our results, showing that a significant increase in serum levels of Docosapentaenoic acid in periodontitis patients with tooth loss and a trend for increased serum levels of Ascorbic acid in periodontitis patients supports the findings from the proband’s reported nutritional intake.

In addition, the overall aim of serum homeostasis needs to be considered. Humans lack Ascorbic acid reservoirs, and thus an excessive intake would not be stored, making a sufficient daily intake still essential. Moreover, positive long-term periodontitis therapy outcomes associated with an adjunctive vitamin supplementation seem to lose their effectiveness compared to short-time adjunctive supplementation and isolated long-term conventional periodontal therapy outcomes [19]. The authors suggest that a number of factors, including a tendency for recalcitrant disease to develop at this stage, could be a reason for the observed reduced improvements at the end of the study phase [19]. Due to the well-established anti-inflammatory properties of vitamin C and positive attributes associated with Ω3FAs, patients themselves could be inclined to consume more vitamin C and DPA-containing groceries to support their periodontal health/healing ability. However, daily vitamin C and omega-3 fatty acid requirement can be altered by stress, smoking, acute infections and chronic diseases. Thus, in periodontitis patients, the daily requirement is increased [6]. Therefore, despite an observed elevated intake of vitamin C and DPA in the periodontitis group compared to controls, their daily required elevated amount of vitamin C and DPA, due to the periodontal inflammatory processes, might still have not been adequately covered. An insufficient nutrient intake consistently impairs innate and adaptive immunity, including phagocytic function, cell-mediated immunity, complement system action, secretory antibody and cytokine production and function [30]. This might explain why periodontitis can occur even with an increased oral intake of vitamin C and DPA and serum levels of DPA in periodontitis patients compared to controls.

It also remains to be discussed whether altered metabolism can lead to impaired absorption or tissue delivery. An investigation demonstrated a positive association between a loss of functional variant of the SLC23A1 vitamin C transporter and periodontitis [31], suggesting that genetic factors might affect periodontitis indirectly through a lack of functional provision of important substrates. Moreover, the elevated consumption of vitamin C in PWL can also be seen as a reflection of the patient’s own increased disease awareness. Thus, patients with tooth loss might better recognize the severity and address more motivated environmental factors, such as their nutrition.

Although the evidence of influencing factors such as urban versus rural lifestyle and different socio-economic factors seems limited, the territorial limitation of the investigated cohort needs to be considered. A fish-rich diet is common in the north of Germany, offering a high source of omega-3 fatty acids, with general anti-inflammatory effect [32] and locally modulating effect on periodontitis or other chronic inflammatory disease, such as rheumatoid arthritis [33,34]. It must be noted that during filling out the questionnaire, subjects could have reflected more critically on their own nutrition or their knowledge about an existing periodontal disease, which might have encouraged them to consume healthier food. Further studies are needed, comparing the oral intake of nutrients with the corresponding systemic but also local concentrations of micronutrients such as vitamin C and DPA, in patients with a clinically and radiologically well-defined time course of periodontal disease, especially by clinicians and dieticians and not only by questionnaires, as carried out in the current investigation. On the other hand, the used questions were in line with the new classification of periodontitis [35,36], identifying severity based on tooth loss, which the patients surely recognized. It should be noted, however, that subjects could suffer from periodontal disease without being aware of it and therefore could have been, even if in part, falsely grouped as controls.

Nutritional needs differ from patient to patient and depend on a variety of factors, including physical condition, age, gender, health and the presence as well as severity of chronic disease such as periodontitis, as investigated. The results of the current investigation indicate that an extended comparison seems reasonable. The food intake in patients with periodontitis could differ in the amount needed between supplement intake and systemic provision. This paradox of micronutrients recognized as anti-inflammatory, being associated with an inflammatory disease needs to be further evaluated in a more comprehensive way, i.e., interventional studies with vitamin C or DPA being applied and examined locally or systemically in periodontitis patients. Possibly, the attempts of periodontitis patients to eat healthier with the goal to improve their dental status might, however, not be automatically mirrored by their serum metabolites availability, still failing to have a positive impact on the periodontitis of the individuals. Increasing efforts for nutrition counseling in periodontitis should gain further importance in the near future, to individualize the periodontal therapy.

## 5. Conclusions

In conclusion, the investigated data of this periodontitis patient cohort and controls demonstrate a relationship between nutritional intake of vitamin C and Docosapentaenoic acid (an omega-3 fatty acid) and its serum metabolite concentrations in periodontal health and disease. A significant increase in serum levels of Docosapentaenoic acid in periodontitis patients with tooth loss and a trend for increased serum levels of Ascorbic acid in periodontitis patients supports the nutritional intake findings. The present results suggest that (I) nutritional behavior might affect the disease process of periodontitis and its therapeutic treatment outcome, and (II) further studies are required to evaluate the paradox of micronutrients recognized as anti-inflammatory, being associated with an inflammatory disease and the complex relationship between dietary intake and bioavailability of micronutrients. This should lead to nutritional counseling in periodontitis patients as a part of an individualized periodontal therapy.

## Figures and Tables

**Figure 1 nutrients-14-01939-f001:**
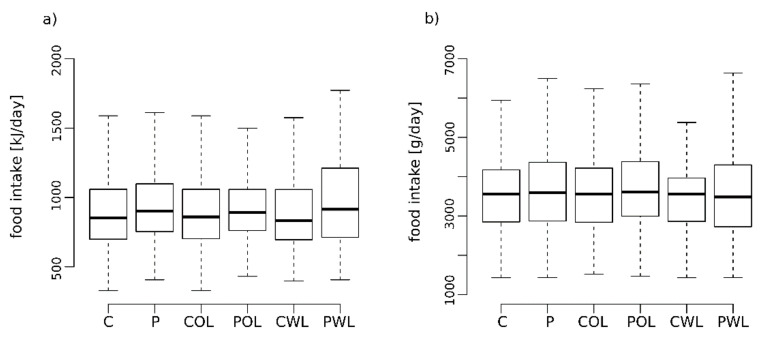
Box whisker plot of daily food intake measured in (**a**) kilojoules and (**b**) grams per day of all controls (C) and periodontitis patients (P), as well as the periodontitis groups without (POL)—and with tooth loss (PWL) with pair matched controls (COL, CWL). Extreme outliers (3xIQR) are not shown in the figure but were included in the analysis. *p*-values for group comparison: (**a**) (C vs. P: *p* = 0.054; COL vs. POL: *p* = 0.23; CWL vs. PWL: *p* = 0.11), (**b**) (C vs. P: *p* = 0.40; COL vs. POL: *p* = 0.26; CWL vs. PWL *p* = 0.86).

**Figure 2 nutrients-14-01939-f002:**
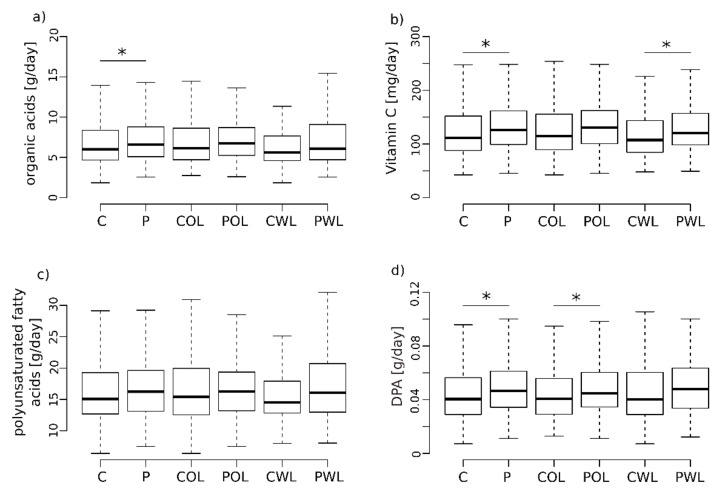
Box whisker plot of organic acids and fatty acids between all controls (C) and periodontitis patients (P), as well as the periodontitis groups without (POL)—and with tooth loss (PWL) with pair matched controls (COL, CWL). Extreme outliers (3xIQR) are not shown in the figure but were included in the analysis. * Significant findings; (**a**) Dietary intake of organic acids (C vs. P: *p* = 0.021 *; COL vs. POL: *p* = 0.132; CWL vs. PWL: *p* = 0.083), (**b**) Dietary intake of vitamin C (C vs. P: *p* = 0.007 *; COL vs. POL: *p* = 0.095; CWL vs. PWL *p* = 0.022 *), (**c**) Intake of polyunsaturated fatty acids (C vs. P: *p* = 0.087; COL vs. POL: *p* = 0.385; CWL vs. PWL *p* = 0.113), (**d**) Intake of Docosapentaenoic acid (DPA) (C vs. P: *p* = 0.0033 *; COL vs. POL: *p* = 0.026 *; CWL vs. PWL *p* = 0.054).

**Figure 3 nutrients-14-01939-f003:**
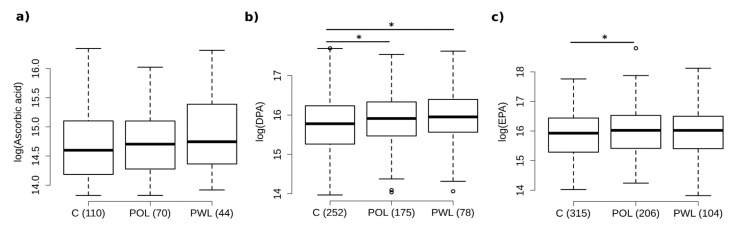
Box whisker plot of Ascorbic acid (**a**), Docosapentaenoic (DPA), (**b**) and Eicosapentaenoic acid (EPA), (**c**). Logistic regression was applied including the matching parameters. Metabolites were log-transformed. Extreme outliers (3xIQR) are not shown in the figure but were included in the analysis. Numbers in brackets are group sizes. * denotes *p*-values < 0.05.

**Table 1 nutrients-14-01939-t001:** Comparing the matching criteria between periodontitis patients and controls. Values are shown as numbers (*n*), percentages (%) or median (IQR). *p*-values were computed using Wilcoxon test or, for binomial data (gender, smoker), chi-square test.

	Periodontitis Patients	Controls	*p*-Value
All periodontitis patients (unstratified) and controls			
Subjects (*n*)	326	326	-
Male (%)	40	40	*p* = 1
Smoker (%)	65	65	*p* = 1
Age (yrs)	59 (50–67)	59 (50–67)	*p* = 0.784
BMI (kg/m^2^)	28.7 (24.9–36.9)	27.5 (24.4–35.5)	*p* = 0.087
Periodontitis without tooth loss and controls			
Subjects (*n*)	213	213	-
Male (%)	38	38	*p* = 1
Smoker (%)	63	63	*p* = 1
Age (yrs)	56 (48–65)	57 (49–65)	*p* = 0.738
BMI (kg/m^2^)	28 (24.7–36.8)	27.5 (24.3–33.8)	*p* = 0.248
Periodontitis with tooth loss and controls			
Subjects (*n*)	113	113	-
Male (%)	46	46	*p* = 1
Smoker (%)	75	75	*p* = 1
Age (yrs)	63 (55–69)	63 (53–70)	*p* = 0.933
BMI (kg/m^2^)	29.9 (25.5–37.8)	27.5 (25.1–35.8)	*p* = 0.158

**Table 2 nutrients-14-01939-t002:** Comparing components of the nutritional intake between controls (C) and periodontitis patients (P) using Wilcoxon test (uncorrected *p*-values are shown), *p*-values in bold are nominally significant and *p*-values marked with an asterisk (*) are robust under multiple testing according to Bonferroni method.

	Component (C vs. P)	*p*-Value		Component (C vs. P)	*p*-Value
fatty acids	* **All saturated fatty acids** *	*p* = 0.086	vitamins	**vitamins**	***p* = 0.020**
	Octadecanoic acid	*p* = 0.057		**fat**	*p* = 0.058
	Octanoic acid	*p* = 0.080		**proteins**	*p* = 0.077
	Hexadecanoic acid	*p* = 0.080		**minerals**	*p* = 0.131
	Dodecanoic acid	*p* = 0.082		**carbohydrates**	*p* = 0.255
	Eicosanoic acid	*p* = 0.105		**fibers**	*p* = 0.342
	Tetradecanoic acid	*p* = 0.144		vitamin C	***p* = 0.007**
	Decanoic acid	*p* = 0.154		vitamin D	***p* = 0.033**
	Butanoic acid	*p* = 0.292		vitamin E	*p* = 0.064
	Heptadecanoic acid	*p* = 0.291		vitamin B5	*p* = 0.080
	Hexanoic acid	*p* = 0.294		vitamin B1	*p* = 0.085
	Pentadecanoic acid	*p* = 0.331		vitamin B6	*p* = 0.088
	Teracosanoic acid	*p* = 0.342		vitamin B9	*p* = 0.148
	Decosanoic acid	*p* = 0.393		vitamin B3	*p* = 0.152
	* **All monounsaturated fatty acids** *	*p* = 0.057		vitamin B7	*p* = 0.197
	Eicosenoic acid	***p* = 0.009**		vitamin K	*p* = 0.199
	Decosenoic acid	***p* = 0.011**		vitamin B12	*p* = 0.232
	Tetracosenic acid	***p* = 0.015**		vitamin B2	*p* = 0.326
	Hexadecenoic acid	*p* = 0.053		vitamin A	*p* = 0.841
	Octadecenoic acid	*p* = 0.061			
	Heptadecenoic acid	*p* = 0.325			
	Tetradecenoic acid	*p* = 0.342			
	Pentadecenoic acid	*p* = 0.344			
	* **All polyunsaturated fatty acids** *	*p* = 0.087			
	Docosapentaenoic acid	***p* = 0.003 ***			
	Docosahexaenoic acid	***p* = 0.008**			
	Eicodonic acid	***p* = 0.018**			
	Eicosatrienoic acid	***p* = 0.018**			
	Eicosatetraenoic acid	***p* = 0.019**			
	Octadecatetraenoic acid	***p* = 0.024**			
	Hexadecadienoic acid	***p* = 0.025**			
	Docosatetraenoic acid	***p* = 0.037**			
	Eicosadienoic acid	***p* = 0.043**			
	Octadecatrienoic acid	*p* = 0.083			
	Octadecadienoic acid	*p* = 0.13			
	Nonadecatrienoic acid	*p* = 0.243			

## Data Availability

The data that support the findings of this study are available on request from the corresponding author (D.M.S.). The data are not publicly available due to privacy or ethical restrictions.

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
