# Peer review of "Vitamin C and Omega-3 Fatty Acid Intake Is Associated with Human Periodontitis—A Nested Case-Control Study"

_nutrients, 2022, doi:10.3390/nu14091939_

Round 1

Reviewer 1 Report

Nice article. No comment.

Author Response

Thank you for your comment. We appreciate your positive feedback.

OUR REPLY: We would like to thank the Associate Editor for coordinating this review and for highlighting the main drawbacks. All suggested points have been considered and addressed. With regards to the Kiel Food Chain Plus (FoCus) cohort, this has now been clarified. Other issues related to our method (including the study design of mixed questionnaire and blood values) and primary outcome have been addressed in response to each of the referees’ comments.

Reviewer 2 Report

The authors have performed very interesting study and the results were quite contrary to the common beliefs. It was generally good narration throughout the manuscript, however, there are some points which should be clarified before the recommendation for the publication.

  1. How did authors determine the periodontitis state? Was it by questionnaire or did clinicians perform radiographic/clinical examinations?
  2. Why was periodontitis included only? Gingivitis and gingival bleeding are also significantly related to the deficiency of Vitamin C. 
  3. Was there any standardization between examiners?
  4. It seems that smoking population is substantially high. Is this normal or did researchers chose this specific ratio?
  5. Although it was stated that "Recruitment, phenotyping of dental status, questionnaire and matching periodontitis
    patients to controls were performed as published by Schulz, Knappe, Graetz et al. [20]." it would be still better to describe briefly how the recruitment was performed. Also the Phenotyping of dental status and Questionnaire might be briefly explained.
  6. It is not clear how the Kiel Food Chain Plus (FoCus) cohort is composed of and what kind of cohort it is. The reader without the background knowledge would be curious regarding what kind of environment this cohort shares.

Author Response

OUR REPLY: We would like to thank the Associate Editor for coordinating this review and for highlighting the main drawbacks. All suggested points have been considered and addressed. With regards to the Kiel Food Chain Plus (FoCus) cohort, this has now been clarified. Other issues related to our method (including the study design of mixed questionnaire and blood values) and primary outcome have been addressed in response to each of the referees’ comments.

How did authors determine the periodontitis state? Was it by questionnaire or did clinicians perform radiographic/clinical examinations?

Thank you for your comment. Phenotyping of the dental status (Self-report periodontal disease information)was performed via questionnaires.A clinical dental examination was not included. The participants were asked the following questions: Has periodontitis (gum disease) been diagnosed by a physician? If yes, in which year has periodontitis been diagnosed or how old have you been. Have you been or are you under dental treatment because of periodontitis. Did you lose teeth due to periodontitis (gum disease with progressed tooth loosening)? Participants were excluded from further analyses when statements were inconsistent. Only subjects who denied all three questions were accepted as potential controls. Selfreported painful gums, bleeding, gums, and loose teeth have previously been validated as surrogate makers of periodontal disease(Eke et al., 2013, Abbood et al., 2016). Adjustments have been made accordingly and can be found in the manuscript, please see page: 2, line: 35-41.

Why was periodontitis included only? Gingivitis and gingival bleeding are also significantly related to the deficiency of Vitamin C. 

Thank you for your comment. The Food Chain Plus (FoCus) cohort was launched in 2011 for population-based research in the so called “metabolic inflammation”. To characterize this novel pathology in a comprehensive manner, data collection included multiple omics layers like phenomics, metabolomics, genomics and metagenomics as well as nutrition profiling, taste perception phenotyping and social network analysis(Muller et al., 2015, Wang et al., 2016). A retrospective analysis of gingivitis and gingival bleeding was not possible due to the questionnaire design and the limited amount of asked questions regarding oral health. The cohort study design initially focused on periodontitis as one of the most common chronic inflammatory diseases worldwide and a crucial player in systemic low-grade, metabolic inflammation.

Was there any standardization between examiners?

Thank you for your comment. As there was no clinical dental examination performed, no standardisation between examiners was done.

It seems that smoking population is substantially high. Is this normal or did researchers chose this specific ratio?

Thank you for your comment. Smoking is considered one of the behavioral risk factors for periodontitis.Numerous studies have shown an association of smoking with poor periodontal status, (Bergström, 1989, Haber and Kent, 1992, Locker, 1992, Haber et al., 1993, Stoltenberg et al., 1993, Jette et al., 1993, Martinez-Canut et al., 1995, Kaldahl et al., 1996, Grossi et al., 1997, Axelsson et al., 1998, Tomar and Asma, 2000, Bergström et al., 2000, Albandar et al., 2000, Susin et al., 2004), as well as a statistically significant increased risk of periodontal disease progression (Machtei et al., 1999, Norderyd et al., 1999, Ogawa et al., 2002). To avoid a potential influence, the smoking history was assigned in a 1 to 1 matching between patients and controls. No specific ratio was formed, nor was there any influence on the number of smokers.

Although it was stated that "Recruitment, phenotyping of dental status, questionnaire and matching periodontitis patients to controls were performed as published by Schulz, Knappe, Graetz et al. [20]." it would be still better to describe briefly how the recruitment was performed. Also the Phenotyping of dental status and Questionnaire might be briefly explained.

Thank you for your comment. Adjustments - in brief - have been made accordingly and can be found in the manuscript, please see page: 2, line: 35-41.

It is not clear how the Kiel Food Chain Plus (FoCus) cohort is composed of and what kind of cohort it is. The reader without the background knowledge would be curious regarding what kind of environment this cohort shares.

Thank you for your comment.

Key Features of the FoCus cohort:

  • The Food Chain Plus (FoCus) cohort was launched in 2011 for population-based research in the so called “metabolic inflammation”. To characterize this novel pathology in a comprehensive manner, data collection included multiple omics layers like phenomics, metabolomics, genomics and metagenomics as well as nutrition profiling, taste perception phenotyping and social network analysis.
  • The Cohort is located in the North of Germany in the Kiel region. According to the Kiel Residents' Registration Office, as of September 30, 2021, a total of 246,518 people were registered in Kiel with sole or main residence.
  • In total, 1803 individuals were recruited at baseline. Baseline data collection took place between 2011 and 2014, including 63.1% of females and 36.9% males with an age range of 18 to 75 years.
  • It is part of a collaborative project funded by the German Federal Ministry of Education and Research.

Adjustments have been made accordingly and can be found in the manuscript, please see page: 2, line: 27-28.

Abbood, H. M., Hinz, J., Cherukara, G. & Macfarlane, T. V. (2016) Validity of Self-Reported Periodontal Disease: A Systematic Review and Meta-Analysis. J Periodontol87,1474-1483. doi:10.1902/jop.2016.160196.

Albandar, J. M., Streckfus, C. F., Adesanya, M. R. & Winn, D. M. (2000) Cigar, pipe, and cigarette smoking as risk factors for periodontal disease and tooth loss. J Periodontol71,1874-1881. doi:10.1902/jop.2000.71.12.1874.

Axelsson, P., Paulander, J. & Lindhe, J. (1998) Relationship between smoking and dental status in 35-, 50-, 65-, and 75-year-old individuals. J Clin Periodontol25,297-305. doi:10.1111/j.1600-051x.1998.tb02444.x.

Bergström, J. (1989) Cigarette smoking as risk factor in chronic periodontal disease. Community Dent Oral Epidemiol17,245-247. doi:10.1111/j.1600-0528.1989.tb00626.x.

Bergström, J., Eliasson, S. & Dock, J. (2000) A 10-year prospective study of tobacco smoking and periodontal health. J Periodontol71,1338-1347. doi:10.1902/jop.2000.71.8.1338.

Eke, P. I., Dye, B. A., Wei, L., Slade, G. D., Thornton-Evans, G. O., Beck, J. D., Taylor, G. W., Borgnakke, W. S., Page, R. C. & Genco, R. J. (2013) Self-reported measures for surveillance of periodontitis. J Dent Res92,1041-1047. doi:10.1177/0022034513505621.

Grossi, S. G., Zambon, J., Machtei, E. E., Schifferle, R., Andreana, S., Genco, R. J., Cummins, D. & Harrap, G. (1997) Effects of smoking and smoking cessation on healing after mechanical periodontal therapy. J Am Dent Assoc128,599-607. doi:10.14219/jada.archive.1997.0259.

Haber, J. & Kent, R. L. (1992) Cigarette smoking in a periodontal practice. J Periodontol63,100-106. doi:10.1902/jop.1992.63.2.100.

Haber, J., Wattles, J., Crowley, M., Mandell, R., Joshipura, K. & Kent, R. L. (1993) Evidence for cigarette smoking as a major risk factor for periodontitis. J Periodontol64,16-23. doi:10.1902/jop.1993.64.1.16.

Jette, A. M., Feldman, H. A. & Tennstedt, S. L. (1993) Tobacco use: a modifiable risk factor for dental disease among the elderly. Am J Public Health83,1271-1276. doi:10.2105/ajph.83.9.1271.

Kaldahl, W. B., Johnson, G. K., Patil, K. D. & Kalkwarf, K. L. (1996) Levels of cigarette consumption and response to periodontal therapy. J Periodontol67,675-681. doi:10.1902/jop.1996.67.7.675.

Locker, D. (1992) Smoking and oral health in older adults. Can J Public Health83,429-432.

Machtei, E. E., Hausmann, E., Dunford, R., Grossi, S., Ho, A., Davis, G., Chandler, J., Zambon, J. & Genco, R. J. (1999) Longitudinal study of predictive factors for periodontal disease and tooth loss. J Clin Periodontol26,374-380. doi:10.1034/j.1600-051x.1999.260607.x.

Martinez-Canut, P., Lorca, A. & Magán, R. (1995) Smoking and periodontal disease severity. J Clin Periodontol22,743-749. doi:10.1111/j.1600-051x.1995.tb00256.x.

Muller, N., Schulte, D. M., Turk, K., Freitag-Wolf, S., Hampe, J., Zeuner, R., Schroder, J. O., Gouni-Berthold, I., Berthold, H. K., Krone, W., Rose-John, S., Schreiber, S. & Laudes, M. (2015) IL-6 blockade by monoclonal antibodies inhibits apolipoprotein (a) expression and lipoprotein (a) synthesis in humans. J Lipid Res56,1034-1042. doi:10.1194/jlr.P052209.

Norderyd, O., Hugoson, A. & Grusovin, G. (1999) Risk of severe periodontal disease in a Swedish adult population. A longitudinal study. J Clin Periodontol26,608-615. doi:10.1034/j.1600-051x.1999.260908.x.

Ogawa, H., Yoshihara, A., Hirotomi, T., Ando, Y. & Miyazaki, H. (2002) Risk factors for periodontal disease progression among elderly people. J Clin Periodontol29,592-597. doi:10.1034/j.1600-051x.2002.290702.x.

Stoltenberg, J. L., Osborn, J. B., Pihlstrom, B. L., Herzberg, M. C., Aeppli, D. M., Wolff, L. F. & Fischer, G. E. (1993) Association between cigarette smoking, bacterial pathogens, and periodontal status. J Periodontol64,1225-1230. doi:10.1902/jop.1993.64.12.1225.

Susin, C., Oppermann, R. V., Haugejorden, O. & Albandar, J. M. (2004) Periodontal attachment loss attributable to cigarette smoking in an urban Brazilian population. J Clin Periodontol31,951-958. doi:10.1111/j.1600-051x.2004.00588.x.

Tomar, S. L. & Asma, S. (2000) Smoking-Attributable Periodontitis in the United States: Findings From NHANES III. J Periodontol71,743-751. doi:10.1902/jop.2000.71.5.743.

Wang, J., Thingholm, L. B., Skieceviciene, J., Rausch, P., Kummen, M., Hov, J. R., Degenhardt, F., Heinsen, F. A., Ruhlemann, M. C., Szymczak, S., Holm, K., Esko, T., Sun, J., Pricop-Jeckstadt, M., Al-Dury, S., Bohov, P., Bethune, J., Sommer, F., Ellinghaus, D., Berge, R. K., Hubenthal, M., Koch, M., Schwarz, K., Rimbach, G., Hubbe, P., Pan, W. H., Sheibani-Tezerji, R., Hasler, R., Rosenstiel, P., D'Amato, M., Cloppenborg-Schmidt, K., Kunzel, S., Laudes, M., Marschall, H. U., Lieb, W., Nothlings, U., Karlsen, T. H., Baines, J. F. & Franke, A. (2016) Genome-wide association analysis identifies variation in vitamin D receptor and other host factors influencing the gut microbiota. Nat Genet48,1396-1406. doi:10.1038/ng.3695.

Reviewer 3 Report

The authors found contradictory results concerning periodontitis and Omega-3-FA in the blood to a lot of very good clinical study I know and performed myself. The amount of Omega-3-fatty acids found in the blood samples is very low and are no indicator for concentrations in the tissues. It is hard to draw conclusions on that.

Author Response

OUR REPLY: We would like to thank the Associate Editor for coordinating this review and for highlighting the main drawbacks. All suggested points have been considered and addressed. With regards to the Kiel Food Chain Plus (FoCus) cohort, this has now been clarified. Other issues related to our method (including the study design of mixed questionnaire and blood values) and primary outcome have been addressed in response to each of the referees’ comments.

The authors found contradictory results concerning periodontitis and Omega-3-FA in the blood to a lot of very good clinical study I know and performed myself. The amount of Omega-3-fatty acids found in the blood samples is very low and are no indicator for concentrations in the tissues. It is hard to draw conclusions on that.

Thank you for your comment and discussion. Nutrition will get more and more in the focus as an inevitable part of medical treatment. With regard to all the possibilities of personalized and precision nutritional medicine, drawing conclusions from oral intake only and disease outcome should be interpreted with care. Due to many more factors i.e. the microbiom, what we eat cannot necessarily be measured in the circulation let alone reflects the concentrations at the site of action. Due to many more omics available, future studies should try to combine the analyses of nutrients in all three compartments – food intake (for nutritional counselling), serum metabolomics (for interpretating blood results) and concentrations in the tissue (for understanding the mechanism). With our paper we would like to contribute to that new approach. However, in our FoCus cohort, we did not have on-site samples available. This should be included in future studies. We are aware of the results of our study comparing other studies. We agree with you that conclusions from blood samples about tissue concentrations can hardly be made. In the manuscript, we therefore discuss the bio-availability in general. In future, we are more than happy to cooperate with you in performing studies to strengthen our new dogma: food intake – metabolomics – onsite action => clinical outcome. We adjusted our manuskript to underline your points; i.e. line 262, 290, 308, 320, 323.

Round 2

Reviewer 3 Report

The manuscript is ok in that form.